# Effects of Substrate Temperature on Nanomechanical Properties of Pulsed Laser Deposited Bi$_2$Te$_3$ Films

Hui-Ping Cheng [1], Phuoc Huu Le [2,*], Le Thi Cam Tuyen [3,*], Sheng-Rui Jian [1,4,5,*], Yu-Chen Chung [1], I-Ju Teng [6], Chih-Ming Lin [7] and Jenh-Yih Juang [8]

[1] Department of Materials Science and Engineering, I-Shou University, Kaohsiung 84001, Taiwan; jasonping87@gmail.com (H.-P.C.); ricky12579@gmail.com (Y.-C.C.)
[2] Department of Physics and Biophysics, Faculty of Basic Sciences, Can Tho University of Medicine and Pharmacy, Can Tho City 94000, Vietnam
[3] Department of Chemical Engineering, College of Engineering Technology, Can Tho University, 3/2 Street, Can Tho City 900000, Vietnam
[4] Department of Fragrance and Cosmetic Science, College of Pharmacy, Kaohsiung Medical University, 100 Shi-Chuan 1st Road, Kaohsiung 80708, Taiwan
[5] Department of Mechanical and Automation Engineering, I-Shou University, Kaohsiung 84001, Taiwan
[6] College of Engineering, National Taiwan University of Science and Technology, Taipei 106335, Taiwan; eru7023@gmail.com
[7] Department of Physics, National Tsing Hua University, Hsinchu 30013, Taiwan; cm_lin@phys.nthu.edu.tw
[8] Department of Electrophysics, National Yang Ming Chiao Tung University, Hsinchu 30010, Taiwan; jyjuang@nycu.edu.tw
* Correspondence: lhuuphuoc@ctump.edu.vn (P.H.L.); ltctuyen@ctu.edu.vn (L.T.C.T.); srjian@gmail.com (S.-R.J.); Tel.: +886-7-6577711-3130 (S.-R.J.)

**Abstract:** The correlations among microstructure, surface morphology, hardness, and elastic modulus of Bi$_2$Te$_3$ thin films deposited on *c*-plane sapphire substrates by pulsed laser deposition are investigated. X-ray diffraction (XRD) and transmission electron microscopy are used to characterize the microstructures of the Bi$_2$Te$_3$ thin films. The XRD analyses revealed that the Bi$_2$Te$_3$ thin films were highly (00*l*)-oriented and exhibited progressively improved crystallinity when the substrate temperature ($T_S$) increased. The hardness and elastic modulus of the Bi$_2$Te$_3$ thin films determined by nanoindentation operated with the continuous contact stiffness measurement (CSM) mode are both substantially larger than those reported for bulk samples, albeit both decrease monotonically with increasing crystallite size and follow the Hall—Petch relation closely. Moreover, the Berkovich nanoindentation-induced crack exhibited trans-granular cracking behaviors for all films investigated. The fracture toughness was significantly higher for films deposited at the lower $T_S$; meanwhile, the fracture energy was almost the same when the crystallite size was suppressed, which indicated a prominent role of grain boundary in governing the deformation characteristics of the present Bi$_2$Te$_3$ films.

**Keywords:** Bi$_2$Te$_3$ thin films; XRD; SEM; nanoindentation; pop-in; hardness

## 1. Introduction

Bismuth telluride, Bi$_2$Te$_3$, is a 3D topological insulator and an excellent thermoelectric (TE) material that works well at room temperature [1,2]. TE materials are of interest for heat pump and power generator applications. The performance of TE materials is quantified by a dimensionless figure of merit (*ZT*), expressed as $ZT = S^2 \sigma T / \kappa$, where *S*, σ, κ, and *T* are the Seebeck coefficient, electrical conductivity, thermal conductivity, and absolute temperature, respectively. Generally, thin films can be fabricated by various methods, such as aerosol-assisted chemical vapor deposition [3,4], dip-coating [5], pulsed laser deposition (PLD) [6], etc. PLD offers advantages such as a higher instantaneous deposition rate, relatively high reproducibility, and low costs.



In addition to improving the properties of TE materials, mechanical characterizations are of critical importance when the reliability of TE devices is concerned [2,7,8]. For example, the performances of TE devices can be significantly degraded due to contact loading during operation. In addition, inhomogeneous thermal expansion may occur in TE generators because they are regularly subjected to the cyclic temperature gradient during processing. Consequently, the inhomogeneous thermal expansion/contraction induces repetitive expansion/shrinkage and the corresponding stress/strain in TE materials to possibly cause fatigue cracking, performance degradation, and even failure of the TE generators. Therefore, it is important to have a comprehensive understanding of the mechanical properties of TE materials (i.e., $Bi_2Te_3$) to provide vital information for fabricating efficient and endurable $Bi_2Te_3$-based devices.

Nanoindentation has become a widely used technique for extracting prominent mechanical properties, namely the hardness and elastic modulus, as well as to unveil the dislocation-mediated plastic deformation and the fracture behaviors of a wide variety of nanostructures [9–11] and oxide thin films [12–14]. The relationship between the microstructure and nanomechanical characterizations of the $Bi_2Te_3/Al_2O_3$ (001) thin films deposited at the various $T_S$ by means of PLD is systematically investigated in this study. $Al_2O_3$ (001) is used because it is an insulating–popular substrate and has moderate lattice mismatch between $Bi_2Te_3$ and $Al_2O_3$ (8.7%). It is found that $T_S$ evidently introduces drastic modification in the film's microstructure, crystallinity, and crystallite size, which in turn manipulates the mechanical characterizations, such as hardness, elastic modulus, fracture toughness, and fracture energy of the $Bi_2Te_3$ films.

## 2. Materials and Methods

In this study, a 99.99%-pure tellurium-excessive target ($Bi_2Te_8$) was used to deposit the $Bi_2Te_3$ films on $Al_2O_3$ (001) substrates by PLD. The reason for choosing the tellurium-excessive target was to overcome the issues of high-doping carriers in the Te-deficient non-stoichiometric $Bi_2Te_3$ phase as well as to avoid the formation of unwanted phases. A KrF excimer pulsed laser was used to ablate the target. The energy density and repetition rate of the laser pulses were 5.7 $J/cm^2$ and 10 Hz, respectively. The vacuum chamber was evacuated to a base pressure of $2 \times 10^{-6}$ Torr. Prior to loading into the chamber, the $Al_2O_3$ (001) substrates with a size of 4 mm × 4 mm were sequentially ultrasonically cleaned in acetone, methanol, and deionized water baths for 30 min. Then, helium gas was introduced into the chamber, with the pressure being maintained at 200 mTorr throughout the entire deposition process. The number of laser pulses was 15,000, and the film thickness was 1154–1428 nm. The substrate temperature ($T_S$) was controlled at 225 °C, 250 °C, and 300 °C. We selected these $T_S$ because they were known as the suitable temperatures for growing high-quality $Bi_2Te_3$ films with excellent thermoelectric properties [6].

The crystal structure of the $Bi_2Te_3$ thin films was determined by X-ray diffraction (XRD; Bruker D2, Billerica, MA, USA) using Cu K$\alpha$ radiation (wavelength of 1.5406 Å) in the θ–2θ configuration. The surface morphology and thickness of the films were examined using a field-emission scanning electron microscope (SEM, JEOL JSM-6500, Pleasanton, CA, USA) at an applied voltage of 15 kV, working distance of 10.5 mm, and magnification of 30,000. The composition of the films was analyzed using an Oxford energy-dispersive X-ray spectroscope (EDS) attached to the SEM. For EDS analyses, the accelerating voltage of the electron beam was set at 15 kV, and the dead time and collection time were 22–30% and 60 s, respectively. Digital images from a high-resolution transmission electron microscope (HRTEM; Tecnai F20, ThermoFisher, Waltham, MA, USA) operated at 200 kV were recorded using a Gatan 2 k × 2 k CCD camera system to obtain detailed film-structure information. The HRTEM specimens were prepared using a standard mechanical thinning and Ar ion milling procedure.

An MTS NanoXP® system (MTS Corporation, Nano Instruments Innovation Center, Oak Ridge, TN, USA) with a load force resolution of 50 nN and a displacement resolution of 0.1 nm was used for conducting the nanoindentation tests at room temperature. The

indentation depth of the Berkovich diamond indenter was 55 nm, and the strain rate varied from 0.01 to 1 s$^{-1}$. Additional harmonic modulation was superimposed simultaneously onto the indenter when continuous stiffness measurements (CSM) were performed [15]. The modulation amplitude and frequency were set at 2 nm and 45 Hz, respectively. Special care was taken to ensure that the thermal drift was less than 0.01 nm/s before each test was conducted. We performed at least 20 indents for each sample to ensure the statistical significance of the results.

The projected contact area between the indenter tip and films surface, $A_p$, and the maximum indentation loading, $P_m$, are used to define the hardness as $H = P_m/A_p$. The Berkovich indenter tip, $A_p$, and the contact depth, $h_c$, are correlated as $A_p = 24.56h_c^2$. Following Sneddon's analysis [16], the elastic modulus of the film $(E_f)$ is given by $S_c = 2\beta E_r(\sqrt{A_p}/\sqrt{\pi})$, where $S_c$ is the contact stiffness of the thin film and $\beta$ is a geometric constant, with $\beta \approx 1$ for the Berkovich indenter tip. The reduced elastic modulus $(E_r)$ is further utilized to determine $E_f$ using $1/E_r = [(1 - v_f^2)/E_f + (1 - v_i^2)/E_i]$, with $v$ being the Poisson's ratio and the subscripts, $f$ and $i$, denoting the parameters for the film and the indenter tip, respectively. For the diamond indenter tip used here, $E_i = 1141$ GPa and $v_i = 0.07$ [17], and $v_f = 0.25$ is assumed.

## 3. Results and Discussion

Figure 1a shows the XRD patterns of films, which can be unambiguously indexed as (003), (006), and (0015) of the rhombohedral Bi$_2$Te$_3$ phase (JCPDS card No. 82-0358). It is worth noting that the intensity ratios between the (0 0 6) and (1 0 16) diffraction peaks were 6.7, 7.8, and 20.2 for films deposited at 225 °C, 250 °C, and 300 °C, respectively, which were all substantially higher than the value of 1 obtained from the standard powder diffraction data file (JCPDS card No. 82-0358), confirming the highly (00$l$)-oriented characteristic of the present Bi$_2$Te$_3$ films. In addition, the preferred in-plane orientation was not found by the XRD phi-scan. Therefore, the Bi$_2$Te$_3$ films are polycrystalline and highly c-axis-oriented (or textured films). Noticeably, the crystal structure of Bi$_2$Te$_3$ is rhombohedral with a space group $D_{3d}^5$ ($R\bar{3}m$), which can be represented by a hexagonal primitive cell consisting of three quintuple-layers (QL). Each QL is about 1 nm-thick with 5-atomic-layer stacking in sequence, namely $-(Te^{(1)}-Bi-Te^{(2)}-Bi-Te^{(1)})-$, along the $c$-axis, as depicted schematically in Figure 1b. The bonding between the QLs is the Van der Waals (VdW) Te$^{(1)}$–Te$^{(1)}$ bond, which is significantly weaker than the ionic–covalent Bi-Te bonds within the QLs [6,18].

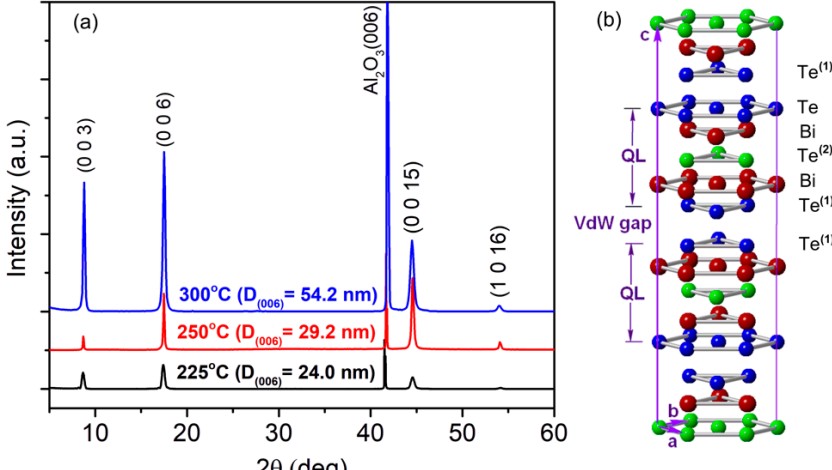

**Figure 1.** (**a**) XRD patterns of the Bi$_2$Te$_3$ thin films deposited on the $c$-plane sapphire substrates at the various $T_S$ of 225 °C, 250 °C, and 300 °C, respectively. D$_{(006)}$ is the calculated grain size of the films using the Scherrer equation and Bi$_2$Te$_3$ (006) peaks. (**b**) The schematics depicts the crystal structure of Bi$_2$Te$_3$.

It is also evident from Figure 1a that the diffraction peaks exhibit higher intensity and narrower width with increasing $T_S$, indicating that the crystallinity of the films is progressively improved when $T_S$ increases. We used Scherrer's formula [19], $D_s = 0.9\lambda/(\beta\cos\theta)$, to estimate the crystallite size ($D_s$) of the $Bi_2Te_3$ thin films, wherein $\lambda$, $\beta$, and $\theta$ are the X-ray wavelength, the full-width-at-half-maximum (FWHM) of the selected diffraction peak, and the corresponding Bragg diffraction angle, respectively. Here, we chose the (006) peak to calculate crystallite sizes, and the sizes are 24.0 nm, 29.2 nm, and 54.2 nm for the films deposited at $T_S$ of 225 °C, 250 °C, and 300 °C, respectively. These results clearly indicate that $T_S$ imposes a marked effect on the crystallite size. This observation can be understood as the following. At a lower $T_S$, due to the higher extent of supersaturation and the lower surface diffusion rate, the reduced critical size and the nucleation energy barrier for the nuclei are reduced, which leads to an increased number of nucleation sites and an eventually smaller grain size [6].

Furthermore, by using the Williamson—Hall (WH) equation [20,21], the effects of the crystalline size-induced broadening and strain-induced broadening of XRD results for the $Bi_2Te_3$ thin films at various $T_S$ can be determined. The WH equation is as follows:

$$\beta\cos\theta = (0.9\lambda/D_{WH}) + 4\varepsilon\cdot\sin\theta \tag{1}$$

where $\varepsilon$ is the microstrain. The WH equation represents a straight line between $4\sin\theta$ (*X*-axis) and $\beta\cos\theta$ (*Y*-axis). The slope of the line gives the value of the microstrain. Both the crystalline size ($D_{WH}$) and microstrain ($\varepsilon$) contribute to the broadening of the XRD spectra of the $Bi_2Te_3$ thin films and are listed in Table 1. It is noted that the incorporation of the microstrain effect is attributed to the different values of the crystalline size significantly. The values of the crystalline size of the $T_S$-treated $Bi_2Te_3$ thin films calculated using Scherrer's equation are noticeably smaller than those calculated using the WH equation that gave rise to the influence of the microstrain on the XRD results. Furthermore, both the values of $D_s$ and $D_{WH}$ are enlarged, indicating that the overall crystallinity of the $Bi_2Te_3$ thin films was remarkably improved by the increased $T_S$. The $\varepsilon$ increases monotonically from 0.21% to 0.29% with increasing $T_S$ from 225 °C to 300 °C, which can be attributed to the newly created interfaces associated with the $T_S$-dependent crystallite size and evolving grain shapes.

**Table 1.** The structural and mechanical properties of the $Bi_2Te_3$ thin films.

| $Bi_2Te_3$ | $D_s$ (nm) | $D_{WH}$ (nm) | $\varepsilon$ (%) | $H$ (GPa) | $E_f$ (GPa) | $\tau_c$ (GPa) | $K_c$ (MPa·m$^{1/2}$) | $G_c$ (Jm$^{-2}$) |
|---|---|---|---|---|---|---|---|---|
| Bulk [22] | — | — | — | 1.6 ± 0.2 | 32.4 ± 2.9 | — | — | — |
| Thin films [23] (helium gas pressure) $2 \times 10^{-5}$–$2 \times 10^{-3}$ Torr | 11–20 | — | — | 2.9–4.0 | 106.3–127.5 | 0.9–1.3 | — | — |
| Thin films [#] $T_S$ = 225 °C | 24.0 | 32.6 | 0.21 | 5.2 ± 0.3 | 125.2 ± 6.9 | 2.2 | 1.42 | 0.15 |
| Thin films [#] $T_S$ = 250 °C | 29.2 | 40.2 | 0.24 | 4.0 ± 0.1 | 98.3 ± 2.1 | 1.4 | 1.21 | 0.14 |
| Thin films [#] $T_S$ = 300 °C | 54.2 | 62.7 | 0.29 | 3.4 ± 0.2 | 62.5 ± 1.4 | 1.0 | 0.88 | 0.12 |

[#]: this work.

Figure 2a shows a low-magnification TEM image $Bi_2Te_3$ film deposited at the $T_S$ of 300 °C, in which some crystallites are presented by the dotted areas. It can be seen that the crystallite sizes are approximately 50–60 nm, agreeing well with the estimated crystallite size obtained from the XRD results. Moreover, an HRTEM image of a crystallite clearly presents the projected periods of 0.51 nm along the c-axis (Figure 2b), which corresponds to the lattice spacing of the $Bi_2Te_3$ (006) planes. These TEM results are further confirmed by the crystallite sizes and gain insight into the crystal structure of the textured $Bi_2Te_3$ films in this study.

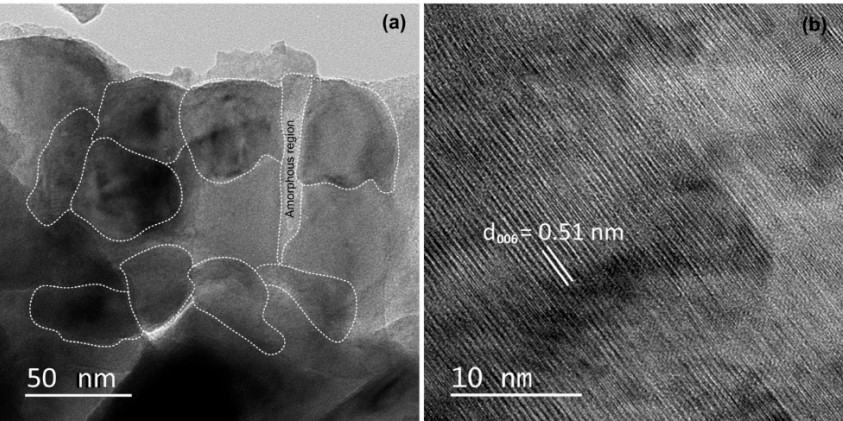

**Figure 2.** TEM images of the $Bi_2Te_3$ film deposited at the $T_S$ of 300 °C. The dotted areas in (**a**) are used to guide the eyes of some crystallites; (**b**) the magnified HRTEM image obtained from (**a**).

Figure 3 depicts the SEM photographs, revealing the surface morphology of the $Bi_2Te_3$ films grown at different $T_S$ from 225 to 300 °C. The SEM images show that all films exhibit the typical polygonal granular morphology of a polycrystalline microstructure, and the grain size increases with increasing $T_S$. This result is in line with the $T_S$-dependent crystallite size tendency obtained from the XRD results (Figure 1). It is noted that the grain size observed from the SEM images shown in Figure 3 appears to be larger than the crystallite size estimated using the XRD data or observed directly from the HRTEM images because a "grain" in the SEM image may be composed of agglomerated grains and/or even include amorphous regions [24]. In addition, the EDS results confirm that all films have stoichiometry very close to that of the $Bi_2Te_3$ phase, which is necessary for obtaining the pure $Bi_2Te_3$ phase for all the investigated films. Noticeably, the Te composition reduces slightly from 60.12 at.% for the 225 °C film to 59.48 at.% for the 300 °C film. This could be due to the re-evaporation of Te from the heated substrates being much faster than that of Bi at elevated temperatures. The cross-sectional SEM images in Figure 3 show that the films had a thickness in the 1154–1428 nm range and layered structures.

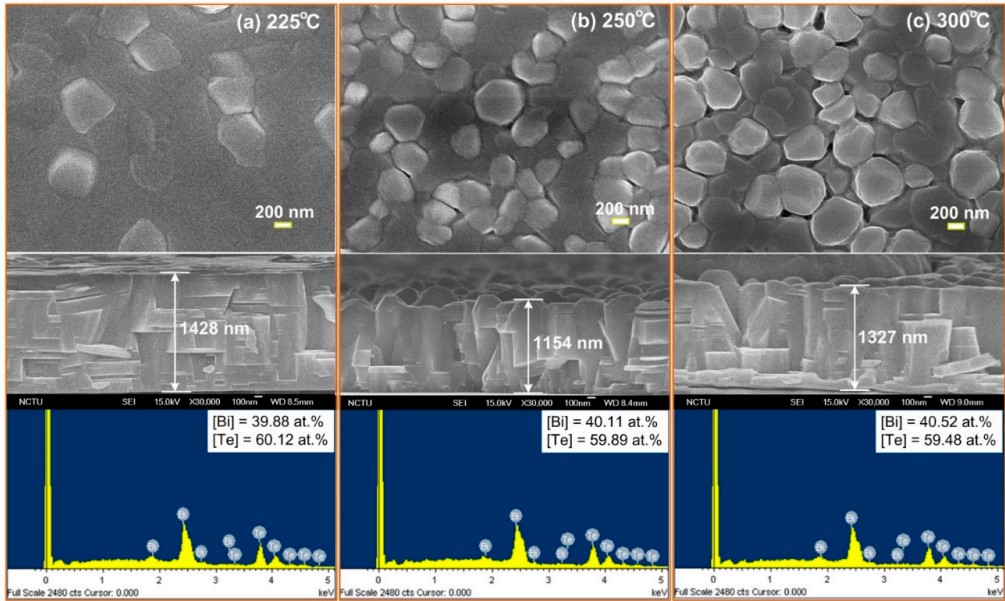

**Figure 3.** SEM images (upper: top view and middle: cross section) and EDS spectra of the $Bi_2Te_3$ thin films deposited at various substrate temperatures of 225, 250, and 300 °C.

As discussed above, the $T_S$ evidently showed significant effects on the film microstructures. The next prominent question is how it affects the nanomechanical properties of the films. Figure 4 shows the typical load—depth curves (*Ph*-curves) obtained from the nanoindentation CSM measurements on the $Bi_2Te_3$ films at various $T_S$ of 225 °C, 250 °C, and 300 °C. We kept the total penetration depth to within approximately 55 nm, which is well below the 30% criterion (film thickness of 1154–1428 nm) suggested by Li et al. [25], in which the indentation depth should never exceed 30% of the films' thickness or the dimension of the nanostructures to avoid any complications from the substrate.

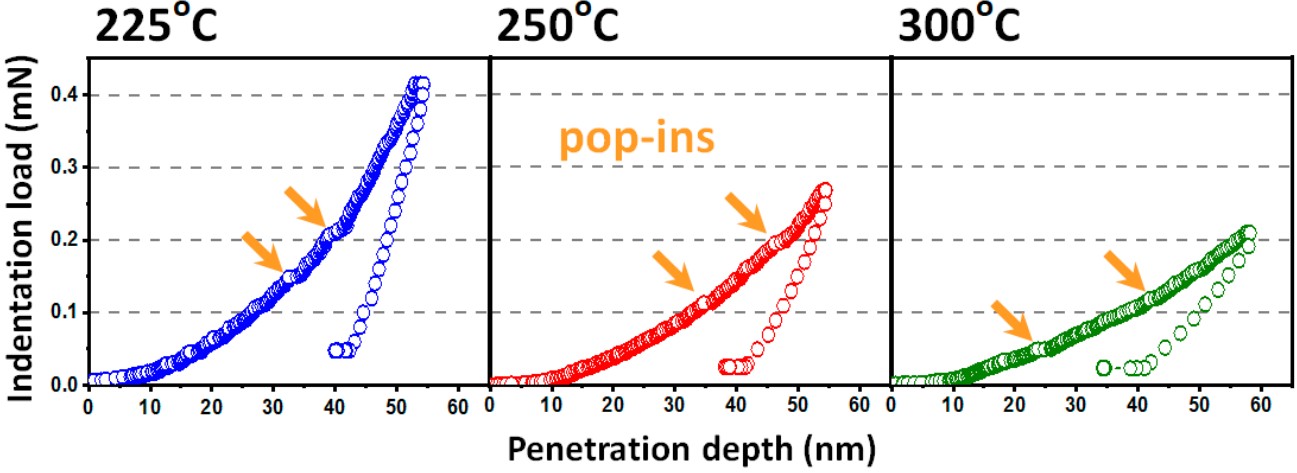

**Figure 4.** CSM nanoindentation *Ph*-curves of the $Bi_2Te_3$ thin films deposited on the *c*-plane sapphire substrates at the various $T_S$ from 225 to 300 °C.

The hardness and Young's modulus of the $Bi_2Te_3$ thin films are directly determined by the *Ph*-curves obtained from CSM measurements following the Oliver and Pharr method [17]. The results are shown in Figure 5. Briefly, the values of hardness (*H*) of the $Bi_2Te_3$ films are 5.2 ± 0.3, 4.0 ± 0.1, and 3.4 ± 0.2 GPa for the films deposited at 225 °C, 250 °C, and 300 °C, respectively. Similarly, the values of Young's modulus ($E_f$) are 125.2 ± 6.9, 98.3 ± 2.1, and 62.5 ± 1.4 GPa for the $Bi_2Te_3$ films grown at 225 °C, 250 °C, and 300 °C, respectively. This means that both the values of *H* and $E_f$ of the $Bi_2Te_3$ thin films monotonically decrease with increasing $T_S$. Noticeably, the *H* and $E_f$ of the present $Bi_2Te_3$ thin films are significantly larger than that reported for the bulk $Bi_2Te_3$ (*H* = 1.6 ± 0.2 GPa and *E* = 32.4 ± 2.9 GPa) [22]. The reason for the apparent discrepancy is not clear at present. Nevertheless, by comparing the results of the films in this study with that reported in Ref. [23], the stoichiometric levels and crystallite orientation of the films may have intimate correlations with the *H* and $E_f$ results. We found that the *H* and $E_f$ values of the close stoichiometric $Bi_2Te_3$ films grown using the $Bi_2Te_8$ target are significantly larger than those of the Te-deficient $Bi_2Te_3$ films grown using the $Bi_2Te_3$ target [23]. Similar behaviors have also been reported in the $Bi_2Se_3$ thin films [26]. Notably, for the present $Bi_2Te_3$ films, the intensity of the (006)-diffraction peak is dominantly higher than that of the (0015)-diffraction peak, whereas the $Bi_2Te_3$ films in Ref. [23] showed reversed behavior for the intensity of the two peaks, presumably due to the different experimental conditions. We found that the mechanical properties of the PLD-grown $Bi_2Te_3$ films on the $Al_2O_3$ (001) substrates substantially were enhanced when they were grown using a Te-rich target (e.g., $Bi_2Te_8$) under a relatively high helium gas (e.g., 200 mTorr) pressure and at a moderately low $T_S$ (e.g., 225 °C). Additionally, compared with the previous studies [22,23], the structural and mechanical properties of the $Bi_2Te_3$ thin films are listed in Table 1.

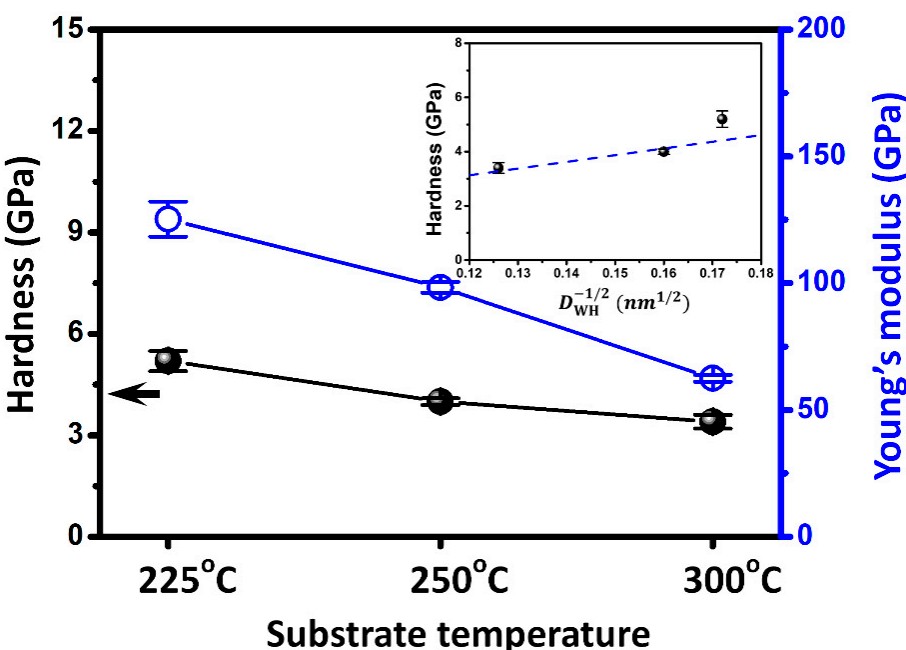

**Figure 5.** The hardness as a function of grain size for the Bi$_2$Te$_3$ thin films. The dash lines are the fitting using the Hall—Petch equation.

On the other hand, as displayed in the inset of Figure 5, the hardness of the films as a function of the crystallite size can be described satisfactorily by the empirical Hall—Petch relation [27], where $H(D_{WH}) = H_0 + kD_{WH}^{-1/2}$ ($H_0$ is the lattice friction stress and $k$ is the Hall-Petch constant). The fitting yields $H(D_{WH}) = 24.7D_{WH}^{-1/2} + 0.2$. Since the dislocation motion is recognized to play the primary role in giving rise to the phenomena describable by the Hall—Petch relation, it can also explain the pop-ins in the *Ph*-curves for the Bi$_2$Te$_3$ thin films by linking the observed pop-ins event to the abrupt plastic flow associated with the massive dislocation activities during nanoindentation (as shown in Figure 4). As is evident from the above results, higher $T_S$ apparently has led to a microstructure with larger crystallite size and better crystallinity for the Bi$_2$Te$_3$ thin films associated with the increased $T_S$ that could reduce the capability of hindering the dislocation movement, hence leading to decreases in the $H$ and $E_f$ values.

As indicated by the arrows shown in Figures 4 and 6, along the loading segment of the *Ph*-curves, clear discontinuities reflecting the pop-ins phenomena are observed. Such behavior, in fact, has been ubiquitously observed in single crystal [22] and thin films [23] of Bi$_2$Te$_3$, when similar nanoindentation tests were undertaken. The fact that it occurs in a vast variety of loading segments associated with a wide range of corresponding strain rates during the test indicates that the phenomena, especially the first pop-in event, are not activated thermally. Instead, the phenomena are often explained in terms of dislocation nucleation and/or propagation [28,29], or development of induced micro-cracks [30,31] during nanoindentation. The possibility of a phase pressure-induced transition, however, can be ruled out. Due to the in situ high pressure XRD experiments carried out on Bi$_2$Te$_3$ [32–34], the magnitude of the applied pressure required to induce the phase transitions is orders of magnitude higher than the apparent room-temperature hardness obtained for the present Bi$_2$Te$_3$ films. Moreover, the absence of "pop-out" discontinuities along the unloading segment of the *Ph*-curves (Figures 4 and 6) also support that, unlike that observed in nanoindented Si [35,36], the phase transition is not involved here. Consequently, we believe that the predominant deformation mechanism prevailing in the present case must mainly associate with the nucleation and subsequent propagation of dislocations.

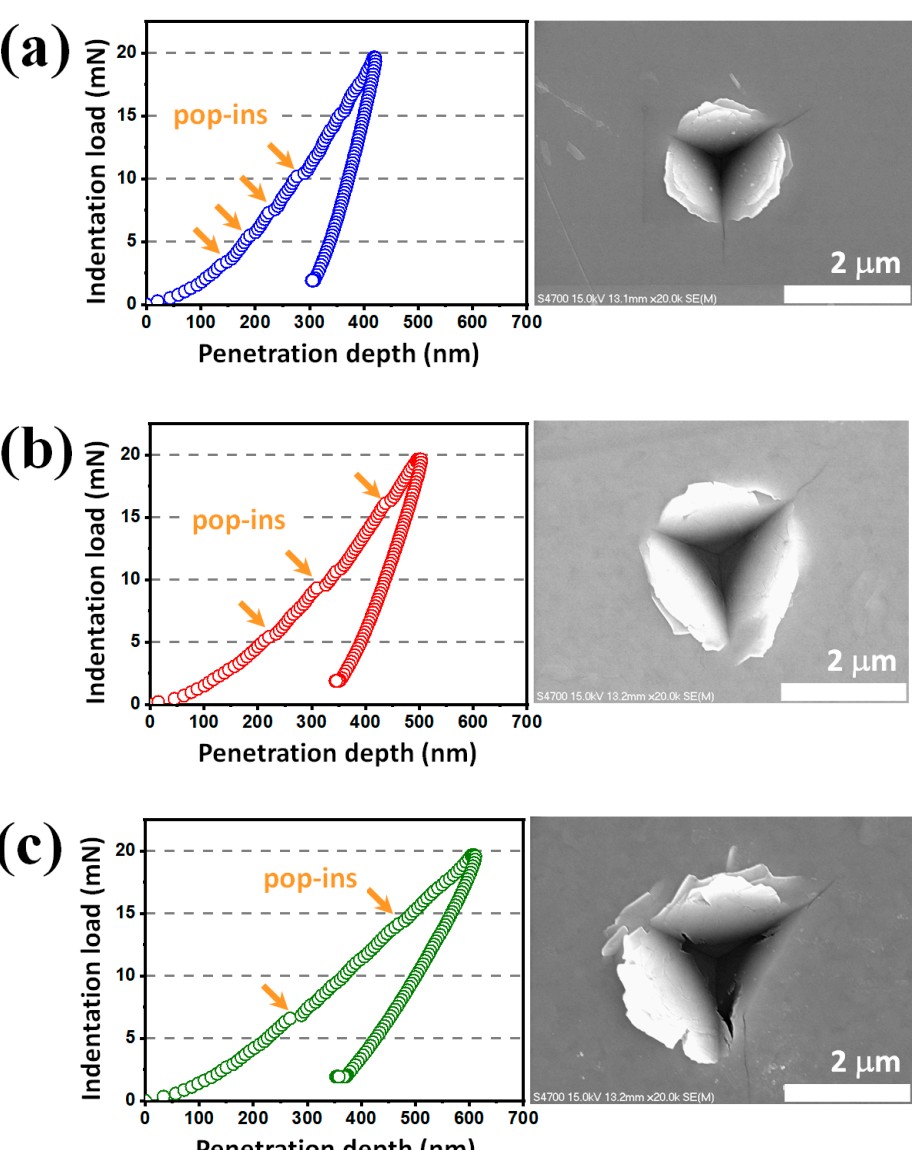

**Figure 6.** Berkovich nanoindentation on the Bi$_2$Te$_3$ thin films at various $T_S$: (**a**) 225 °C, (**b**) 250 °C, and (**c**) 300 °C. The corresponding SEM indentations are displayed to the right of the load-depth curves.

For the context of the dislocation activity-mediated scenario, the first "pop-in" event can be attributed to the transition of deformation behaviors. Namely, the pop-in is the onset of plasticity, reflecting the indentation load at which the system switches from elastic to plastic deformation due to the movement of dislocations. Based on the above discussion, one can further calculate the corresponding critical shear stress ($\tau_c$), at which the dislocation movement is initiated using the following equation: $\tau_c = (0.31/\pi)\left[6P_c(E_r/R)^2\right]^{1/3}$ [37], where $P_c$ is the load at which the load-depth discontinuity occurs, $R$ is the radius of the tip of nanoindenter, and $E_r$ is the reduced elastic modulus defined in the Materials and Method section. The obtained values for $\tau_c$ are approximately 2.2, 1.4, and 1.0 GPa for the Bi$_2$Te$_3$ films deposited at the $T_S$ of 225, 250, and 300 °C, respectively. Alternatively, $\tau_c$ may also be regarded as the stress responsible for massive homogeneous nucleation of the dislocations within the region deformed underneath the tip.

Figure 6 shows the phenomena of Berkovich nanoindentation-induced cracking and the pile-up along the corners and edges of the residual indent clearly. Fracture toughness ($K_c$) is another prominent mechanical property of materials in nanoindentation, which

can be determined by [38] $K_c = \alpha \cdot (P_m/c^{3/2}) \cdot (E_f/H)^{1/2}$, where $\alpha$ is an empirical constant depending solely on the geometry of the indenter, which is taken to be 0.016 for the Berkovich indenter, and $c$ is the trace length of the radial crack appearing on the material surface at a maximum indentation loading ($P_m$) of 20 mN. The $K_c$ values of the $Bi_2Te_3$ thin films thus obtained are 1.42, 1.21, and 0.88 MPa·m$^{1/2}$ for films deposited at the $T_S$ of 225 °C, 250 °C, and 300 °C, respectively. Moreover, the fracture energy ($G_c$) of the $Bi_2Te_3$ thin films is estimated based on the elastic modulus and fracture toughness using the equation [39] $G_c = K_c^2 \cdot (1 - v^2/E_f)$, where the $G_c$ values of the $Bi_2Te_3$ thin films are 0.15, 0.14, and 0.12 Jm$^{-2}$ for the films deposited at the $T_S$ of 225 °C, 250 °C, and 300 °C, respectively. The values of Kc and Gc of the $Bi_2Te_3$ thin films are also listed in Table 1. Accordingly, as is evident from the SEM photographs shown in Figure 3, the cracks propagate in a straight line and exhibit a trans-granular cracking behavior, which confirms that the grain boundaries have effectively obstructed the inter-granular crack propagation.

It is worth noting that the $Bi_2Te_3$ film grown at the $T_S$ of 225 °C can be considered the optimal film because of its excellent mechanical properties of $H = 5.2 \pm 0.3$ GPa and $E_f = 125.2 \pm 6.9$ GPa for thermoelectric applications [40].

### 4. Conclusions

In summary, the $Bi_2Te_3$ thin films were grown on *c*-plane sapphire substrates at various $T_S$ from 225 to 300 °C under a helium ambient pressure of 200 mTorr using a $Bi_2Te_8$ target. The $T_S$ dependence of the structural, morphological, compositional, and nanomechanical properties of the $Bi_2Te_3$ films was systematically studied using XRD, TEM, SEM, EDS, and nanoindentation methods. As a result, all the films exhibited the $Bi_2Te_3$ phase, highly *c*-axis preferred orientation, granular morphology, and good stoichiometry. Moreover, the crystallite size of the films monotonically increased with increasing $T_S$ from 225 to 300 °C. The hardness (Young's modulus) of the $Bi_2Te_3$ thin films decreased from 5.2 GPa (125.2 GPa) to 3.4 GPa (62.5 GPa) when $T_S$ increased from 225 to 300 °C. The $T_S$-dependent hardness and Young's modulus is associated with the variation in crystallite size, which can be explained by the dislocation-mediated mechanism underlying the Hall—Petch relation. The calculated values of $K_c$ and $G_c$ of the $Bi_2Te_3$ thin films were in the ranges of 0.88–1.42 MPa·m$^{1/2}$ and 0.12–0.15 Jm$^{-2}$, and their values were systematically decreased with increasing $T_S$.

**Author Contributions:** H.-P.C., P.H.L., L.T.C.T. and Y.-C.C. contributed to the experiments and analyses. P.H.L., L.T.C.T., S.-R.J., I.-J.T., C.-M.L. and J.-Y.J. contributed to the discussion on materials characterizations. P.H.L., L.T.C.T., S.-R.J. and J.-Y.J. designed the project of experiments and drafted the manuscript. All authors have read and agreed to the published version of the manuscript.

**Funding:** The research was funded by the Ministry of Science and Technology, Taiwan, under Contract Nos. MOST 110-2221-E-214-013, MOST 109-2112-M-009-014-MY2, and Vietnam National Foundation for Science and Technology Development (NAFOSTED) under Grant No. 103.02-2019.374.

**Institutional Review Board Statement:** Not applicable.

**Informed Consent Statement:** Not applicable.

**Data Availability Statement:** Not applicable.

**Acknowledgments:** We acknowledge the support of time and facilities from Can Tho University of Medicine and Pharmacy for this study, and thank the MANALAB at ISU.

**Conflicts of Interest:** The authors declare no conflict of interest.

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
