# Peer review of "Effects of Substrate Temperature on Nanomechanical Properties of Pulsed Laser Deposited Bi2Te3 Films"

_coatings, doi:10.3390/coatings12060871_

Round 1

Reviewer 1 Report

Please see the enclosed file.

Author Response

Dear Reviewer 1,

Please kindly find the attached file.

Thanks and Best,

Jian

Reviewer 2 Report

This review concerns the manuscript coatings-1723489 titled "Effects of substrate temperature on nanomechanical properties of textured Bi2Te3 thin films grown by pulsed laser deposition." The paper describes the mechanical properties of Bi2Te3 deposited onto the sapphire. Depositions were performed under different temperature conditions, resulting in various degrees of crystallinity and microstructures. The manuscript is concise and appropriately characterized, although the introduction could be improved. Overall, this reviewer finds the manuscript suitable for publication upon major revisions intended to improve the discussion.

Comments:

1) What is VdW gap in Figure 1? Not mentioned in the text or figure caption.

2) Indeed, Bi2Te3 is c-axis oriented, but authors could explore better their own XRD results. By comparing the relative intensities of their materials with calculated patterns of powder diffraction files, they can discuss preferential crystal growth with applied temperature. Obviously, this would ignore the (006) of sapphire for calculations of relative intensities. Please see the following manuscript to help discuss this comment: Identification of preferentially exposed crystal facets by X-ray diffraction (https://doi.org/10.1039/D0RA00769B).

3) For the crystallite size calculations. Was the data corrected for instrumental line broadening? Your data is straightforward to perform correction because you have the sapphire (006) diffraction plane as a reference. This can be used as the instrumental line broadening.

4) Have the authors tried to use the Williamson-hall method to calculate crystallite sizes and microstrain? Most interesting is that the technique will give you a microstrain. You can correlate the microstrain with your mechanical properties in Figures 3, 4, and 5.

5) Lines 144 and 145. Please remove the sentence since 60.12% and 59.48% are statistically the same for EDS analysis. Thus, the chemical composition at 225 and 300 oC are the same.

6) Reduce the size of dots in Figures 3 and 5. The pop-in events are hidden.

7) Second paragraph of page 5. The discussion might correlate with the physical connectivity of Bi2Te3 grains. That also affects the intensity of the sapphire (006) diffraction plane. The gaps will allow more diffractions to pass through and reach the detector. Comparatively, the presence of an amorphous phase between crystals will impose limitations.

Author Response

Dear Reviewer 2,

Please kindly find the attached file.

Thanks and Best,

Jian

Reviewer 3 Report

File is attached

Author Response

Dear Reviewer 3,

Please kindly find the attached file.

Thanks and Best,

Jian

Round 2

Reviewer 2 Report

This revision concerns the revised manuscript coatings-1723489-peer-review-v2. The authors have provided revisions for all comments but made a mistake that must be fixed before final acceptance.

(1) There is no use in presenting the sum of relative intensities (the 0.94 or 0.98 mentioned). The powder diffraction files (PDF) display information of 0 to 100% or 0 to 1. This referee's intent in providing a reference was to enrich the discussion and not add citations aimlessly. In a PDF, the most intense peak is 100% (or 1), and everything else is presented as relative intensity to that peak. You can use that as a base reference because PDF files do not account for preferential crystal growth.

This was the idea: When crystals orient in a specific direction, the diffraction peaks corresponding to the lattice planes perpendicular to that direction intensify. For example, you have two materials with the same crystal structure and are analyzing the ratio of planes (110) and (200) – I110/I200. Crystal A has a ratio between (110) to (200) of 0.5 and crystal B a 2.5. The normal value presented in the PDF is 1.0. Thus, crystal B has exposed {110} facets, and crystal A has exposed {100} facets. The latter is perpendicular to {110} facets. This example can be found here: https://doi.org/10.1021/ja2002132.

(2) The microstrain increases with temperature because crystals gain shape creating interfaces. These results are good, but it is better to present them as whole numbers. 

Author Response

Dear reviewer #2,

Jian

Reviewer 3 Report

All the queries raised last time, have been addressed properly. The manuscript can be accepted in its present form.

Author Response

Dear reviewer #3,

Jian
